# Interplay Among Gut Microbiota-Derived TMAO, Autonomic Nervous System Dysfunction, and Heart Failure Progression

**DOI:** 10.3390/ijms27010203

**Published:** 2025-12-24

**Authors:** Laura Calvillo, Emilio Vanoli, Fulvio Ferrara, Eugenio Caradonna

**Affiliations:** 1Istituto Auxologico Italiano IRCCS, 20145 Milan, Italy; 2School of Nursing, University of Pavia, 27100 Pavia, Italy; emilio.vanoli@unipv.it; 3Servizio Integrato di Medicina di Laboratorio, Centro Diagnostico Italiano—CDI, 20147 Milan, Italy; fulvio.ferrara@cdi.it; 4Dipartimento Patologia Medica e Anatomia Patologica, Centro Diagnostico Italiano—CDI, 20147 Milan, Italy; eugenio.caradonna@cdi.it

**Keywords:** trimethylamine-N-oxide (TMAO), microbiota, dysbiosis, cardiovascular disease, autonomic nervous system dysfunction

## Abstract

The gut microbiota is crucial for metabolic homeostasis and cardiovascular health. Dysbiosis triggers a gut–brain–heart axis dysfunction: vagal signaling promotes neuroinflammation and cerebral damage, which in turn impairs cardiac function. This bidirectional cycle is further exacerbated by reduced cerebral perfusion. Trimethylamine-N-oxide (TMAO), a metabolite of dietary choline and L-carnitine, acts as a primary mediator in this network. Elevated TMAO levels—resulting from bacterial conversion and hepatic oxidation—are linked to atherosclerosis and heart failure. Mechanistically, TMAO activates the NLRP3 inflammasome, inhibits the SIRT3-SOD2 pathway, and promotes platelet hyperreactivity. Furthermore, it modulates the autonomic nervous system, enhancing sympathetic activity and cardiac arrhythmias. Clinical evidence suggests TMAO is a potent predictor of mortality in HF. While current HF therapies focus on end-organ response (beta-blockers) or humoral pathways (ACE inhibitors), directly targeting the microbiota and TMAO offers a novel therapeutic frontier. Integrating TMAO assessment into risk models and utilizing advanced in vitro gut–brain models will be essential for developing personalized, groundbreaking cardiovascular interventions. Within this framework, the main aim of the present review is to describe how cardiac autonomic control can be directly modulated by the microbiota and its byproducts like TMAO. This latter is a leading target candidate for novel HF prevention and therapy interventions.

## 1. Introduction

Heart failure (HF) is a complex clinical syndrome involving not only hemodynamic and structural abnormalities but also systemic inflammatory, metabolic, and neurohumoral dysregulation. Among emerging factors involved in HF development and progression, the gut microbiota and its metabolite TMAO have gained considerable attention. TMAO, produced through microbial metabolism of dietary precursors and hepatic oxidation, has been linked to endothelial dysfunction, mitochondrial stress, and chronic low-grade inflammation [1,2]. As previously described, TMAO is also produced in the aorta cellular layer and in adipose tissue [3,4,5]. Recent evidence also implicates TMAO in modulating the autonomic nervous system (ANS), contributing to sympathetic hyperactivity—a known driver of HF progression and arrhythmogenesis. Thus, a critical loop exists between gut microbiota ANS balance and function, inflammation, and cardiovascular diseases. Such a detrimental loop becomes strikingly manifest when considering the proactive interactions between these factors in favoring myocardial infarction occurrence and, more so, in post-MI negative remodeling into left ventricular dysfunction and HF. The growing appreciation of this integrated process is specifically highlighted by very recent publications by the American and European Cardiology and Psychiatric Societies [6,7]. Accordingly, this review integrates current evidence on TMAO as a molecular mediator bridging the gut–heart–brain axis, highlighting its influence on autonomic tone and cardiac outcomes. We also discuss the clinical implications of modulating TMAO levels as a therapeutic target for precision cardiology.

## 2. ANS in HF

It is established knowledge that autonomic responses to a cardiac insult (mostly of ischemic origin) drive most of the pathophysiological processes of negative cardiac remodeling, eventually leading to heart failure [8]. In this context, sympathetic activation has long been appreciated as a fundamental compensatory mechanism of the failing heart, specifically in the presence of left ventricular dysfunction; thus, it is a welcome phenomenon and is to be supported.

In this context, the hypothesis of modulating the reflex sympathetic response to the loss of cardiac function was strongly denied by the cardiology community till the early 1990s. The efficacy of beta-blockers in HF with preserved LV function is still under debate [9], while the undisputed evidence of beta-blockers’ efficacy in heart failure with reduced ejection fraction [10] reversed the understanding of HF’s pathophysiology and opened a completely novel approach to its detection and treatment. Within this revolutionary process, the ATRAMI study [11,12] (Autonomic Tone and Reflexes After Myocardial Infarction) addressed the close relationship between autonomic balance after myocardial infarction (MI) and later mortality risk. The time course of cardiac autonomic control remodeling after a first MI and its influence on later left ventricular remodeling leading to progressive left ventricular dysfunction and sudden death was also documented [13]. This evidence is supported by a number of publications highlighting the role of autonomic imbalance in arrhythmogenesis, sudden death, and HF progression [14]. The key message emerging from both experimental and clinical studies can be expressed in a very condensed sentence: too much sympathetic cardiac drive leads to HF and/or death, while adequate vagal inhibitory control produces a longer and better life. This is, in essence, the key message that generated a novel line of research aimed at restoring balanced cardiac control as soon as possible after a cardiac perturbation, specifically if of ischemic origin [10,15].

Relevant to the present review is the role of the inflammatory process in cardiac mechanical and neural remodeling, driven by nerve growth factor (NGF). NGF is the most representative member of the neurotrophin family and plays a pivotal role in regulating neuronal growth and repair [16,17]. Indeed, preclinical models consistently showed that MI causes, within a very short time, a diffuse upregulation of NGF that is more pronounced in the border zone. The subsequent nerve sprouting signal, facilitated by a rise in the levels of NGF in the Left Stellate Ganglion and in serum, triggers an increase in cardiac nerve density in both peri-infarct and non-infarcted areas, likely leading to an increased risk of arrhythmias. Such findings were confirmed to be of clinical relevance by a study that analyzed cardiac tissue from patients suffering lethal arrhythmic events after MI. On the other hand, NGF is expressed by ischemic myocytes and may also increase cell survival during acute myocardial infarction. NGF promotes cardiac repair following myocardial infarction [18].

Thus, similarly to neural sympathetic activity, whose action is important to sustain the ischemic heart but becomes detrimental if it is excessive, NGF expression is protective if balanced but becomes detrimental if overexpressed. The combination of sympathetic sprouting and sympathetic neural activation indeed creates an anatomical and functional condition of catecholamine toxicity, with major detrimental effects on the entire pathophysiology of HF [16].

### Direct Electrical Stimulation of the Vagus Nerve

The cumbersome path of understanding vagal control of cardiac function was very similar to the process that led to our understanding of antiadrenergic therapy in HF. Once more, despite early evidence dating back to the early 20th century showing that the vagus nerve innervates the ventricles, chronic vagal stimulation was never perceived as a feasible and potentially effective intervention for ischemic heart disease and heart failure. After pioneering work by Verrier and Lown [19] documenting the direct effect of vagal stimulation of ventricular electrophysiology, in the early 1990s, vagal stimulation was able to prevent ventricular fibrillation in awake animals, allowing for the study of sudden cardiac death [20]. The further expanded experimental evidence [21] lead to several clinical studies in which direct or reflex (via baroreceptor stimulation) vagal activation improved quality of life by significantly reducing the hospitalization rate in patients with congestive heart failure [22].

Chronic stimulation of the cervical vagus nerve is very effective both in treating depression and epilepsy [23] and in reducing the HF hospitalization rate. Given the fact that 80% of the vagus trunk is constituted by afferent fibers from the gastro-intestinal apparatus, this implies that afferent inputs from the gut can affect central and autonomic nervous system activity. Thus, microbiota function restoration might indeed be a novel approach to HF therapy by blocking pro-inflammatory agents and restoring cardiac autonomic balance.

## 3. Trimethylamine-N-Oxide

Trimethylamine (TMA) is a compound synthesized by the intestinal microbiota from precursors such as choline, carnitine, lecithin, gamma-butyrobetaine, and phosphatidylcholine. These precursors are abundantly present in common dietary sources, including red meat, saltwater fish, cheeses, eggs, dairy products, and, to a lesser extent, fruits, vegetables, and cereals [24]. TMA is synthesized from specific precursors through the action of bacterial enzymes and is subsequently converted to TMAO. The enzymatic pathways involved in this conversion include choline TMA lyase (genes *cutC*, *cutD*), carnitine monooxygenase (genes *cntA*, *cntB*), betaine reductase, and TMAO reductase. Additionally, TMA can be derived from the reduction of dietary TMAO by the intestinal microbiota. In the liver, TMA is oxidized to TMAO by flavin monooxygenase-3 (FMO3), which exhibits 15 times greater activity than flavin monooxygenase-1 (FMO1), primarily expressed in the kidney and intestine. In humans, there are five functional genes encoding FMO, and individual variations in FMO3 expression are associated with genetic and physiological factors [25]. The aorta expresses the TMAO-synthesizing enzyme (FMO3) in multiple vascular cell types (endothelial cells, fibroblasts, macrophages, pericytes, smooth muscle cells) in both mouse (including high-fat diet-fed) and human aortas, as shown by single-cell RNA sequencing and qPCR [26]. Further, recent evidence has demonstrated that with aging, TMAO is produced in adipose tissue [4]. The production of TMAO is lower in men than in women due to the inhibitory action exerted on FMO3 by testosterone [27].

The genetic architecture that regulates TMAO is complex and probably goes beyond the FMO3 [28]. TMAO is a biologically active metabolite that is eliminated by secretion in the proximal part of the renal tubule.

The physiological role of methylamines, including TMAO, is to stabilize protein structures, thereby protecting them from osmotic stress. However, this stabilizing property means that TMAO can also promote the aggregation of non-functional proteins, such as β-Amyloid, under certain conditions [29,30,31]. Moreover, it has been widely demonstrated that high levels of TMAO have toxic effects and lead to the arrest of cellular synthesis [32].

The gut microbiota is predominantly composed of two main phyla: Bacteroidetes and Firmicutes. Bacterial species capable of producing TMA are widely distributed across multiple phyla, including Actinobacteria and Proteobacteria, though they are most abundant among Firmicutes [33,34]. Beyond the commensal flora, several non-commensal bacteria, such as Burkholderia, Campylobacter, Aeromonas, Salmonella, Shigella, and Vibrio, can also produce TMAO [24]. Interestingly, the host microbiota is capable of performing a retroconversion process in which TMAO is reduced back to TMA by the intestinal flora. Once reabsorbed in the intestine, this TMA can then be reconverted to TMAO by hepatic enzymes [35,36]. Elevated circulating TMAO levels have been identified as an independent risk factor for the development of arteriosclerosis and diabetes [37,38]. Furthermore, TMAO plays a significant role in the pathogenesis of neurodegenerative diseases, including Parkinson’s disease and Alzheimer’s syndrome [29].

### TMAO in Heart Failure

Numerous studies have shown that TMAO is associated with an increased risk of heart failure, especially in patients with existing cardiovascular disease [39,40,41]. TMAO can contribute to the development of heart failure by promoting an inflammatory state with elevated values of cytokines such as Interleukins 6 and 1 (IL-6, IL-1), Tumor Necrosis factor-α (TNF-α), oxidative stress, and endothelial dysfunction and by increasing oxygen consumption, which alters cardiac metabolism in patients with heart failure [42]. In patients with acute heart failure, the combination of TMAO and pro-B-type natriuretic peptide (Pro BNP) improves mortality risk stratification at one year [43]. Tang et al. observed that TMAO represents a very important prognostic factor in the progression of heart failure [44]. The BIOSTAT-CHF study (Biology Study to TAilored Treatment) showed that TMAO levels are correlated with a higher mortality rate, regardless of the current therapy [45]. The interconnections between TMAO and heart failure within the framework of the gut–heart axis are gaining increasing significance [46,47]. Heart failure and kidney disease frequently coexist due to the cardiorenal axis. TMAO is increasingly viewed as marker and a cause of clinical worsening of kidney function [48,49], and TMAO impairs renal function, increasing kidney fibrosis [50].

The relationship between TMAO and the autonomic nervous system, which plays a crucial role in cardiovascular diseases, particularly heart failure and hypertension, has been less extensively studied.

Furthermore, the established role of TMAO in the development of atrial and ventricular arrhythmias constitutes an additional significant aspect in the context of heart failure [51,52,53,54]. A comparative table of key clinical studies on TMAO’s prognostic value is provided in Table 1 and Figure 1. Table 1 and Figure 1 summarize the findings and effect estimates of key clinical studies, demonstrating a consistent association between elevated TMAO levels and increased risk of mortality or adverse events in HF patients.

Table 1 summarizes the findings of key clinical studies investigating the prognostic significance of Trimethylamine N-oxide (TMAO) in patients with HF and related cardiovascular conditions.

## 4. Link Among Vagus Nerve, Gut Dysbiosis, and Cardiac Dysfunction

### 4.1. The Cholinergic Anti-Inflammatory Pathway and Gut Dysbiosis

The cholinergic anti-inflammatory pathway (CAP) is a critical neuroimmune circuit that regulates inflammatory responses to maintain both cardiovascular and systemic physiological balance [57]. This pathway functions as an interface between the brain and the immune system. The efferent (outgoing) branch of this reflex, mediated primarily through the vagus nerve and the neurotransmitter acetylcholine, acts directly on immune cells to dampen their inflammatory output. Inflammation control occurs through interaction with α 7-nicotinic acetylcholine receptors expressed on macrophages and on several other immune cells [57,58,59]. The central nervous system utilizes hardwired neural pathways, such as the vagus nerve, to regulate immune function in real time and in a localized manner [60,61]. The vagus nerve extends to visceral organs, including the spleen and liver (major sources of cytokines), positioning this pathway as a master regulator capable of modulating inflammation across multiple organ systems [60,61,62].

Inflammation is fundamentally a vital, normal biological response orchestrated by the immune system to protect the body. Its primary roles are to eliminate the initial cause of cell injury (such as pathogens, irritants, or damaged cells), clear out necrotic cells and tissue damaged from the original insult and the inflammatory process, and initiate tissue repair. However, the efficacy and safety of this response rely on precise regulation. The critical issue arises when this finely tuned process becomes dysregulated. A response that is either too weak or too strong can lead to serious pathological conditions. Given the destructive potential of unchecked inflammation, the body possesses inherent regulatory mechanisms like CAP, one of the most significant. This pathway serves as a crucial, internal brake on the immune system. Its function is to restrain cytokine production, thereby preventing the response from escalating unnecessarily. Dysfunction in this pathway may predispose individuals to excessive inflammatory responses, and clinical and preclinical evidence supports this concept: laboratory animals with experimentally insufficient efferent signaling in the cholinergic anti-inflammatory pathway consistently exhibit robust, exaggerated systemic inflammatory responses [63].

The disruption of this regulatory circuit through gut dysbiosis establishes a pathological cascade with important cardiovascular consequences [64]. Gut dysbiosis reduces vagal tone, impairing its ability to regulate inflammatory responses. Diminished vagal activity results in uncontrolled inflammation that promotes vascular damage and atherosclerosis [65]. Patients with chronic heart failure show congestion of the intestinal wall and edema, with increased intestinal permeability resulting in the translocation of bacteria and their metabolites with potential effects on cardiovascular health [66]. This compromised intestinal barrier allows bacterial products such as lipopolysaccharides and TMA to enter systemic circulation. TMAO and lipopolysaccharides from intestinal microbiota stimulate Nuclear Factor Kappa-Light-Chain-Enhancer of Activated B Cells (NF-κB) signaling, leading to the release of pro-inflammatory cytokines, including TNF-α, IL-6, and IL-1β, which increase myocardial damage and inhibit cardiac repair [67]. The resulting inflammatory state perpetuates both dysbiosis and cardiovascular pathology, creating a self-reinforcing cycle of deterioration.

### 4.2. The Gut–Brain Axis, Dysbiosis, and Neuroinflammation

This observation underscores the systemic importance of the cholinergic anti-inflammatory pathway, which extends far beyond cardiovascular regulation to encompass metabolic, immunological, and neurological homeostasis. The pathway’s unique ability for rapid and targeted immunomodulation makes its mechanisms of action a crucial topic to be explored and fully understood. In fact, the intricate gut–brain axis acts as a bidirectional communication network, utilizing neural pathways with the primary action of the vagus nerve, to connect gut microbes with the central nervous system. A disturbed microbial balance in the gut signals the brain via vagal pathways, triggering neuroinflammatory responses through activation of the Nod-like Receptor Pyrin Domain-containing 3 (NLRP3) inflammasome in resident microglia. This leads to the release of pro-inflammatory cytokines, including IL-1β, which propagate neuroinflammation through multiple downstream mechanisms [68] (Figure 2). These include, for instance, impaired blood–brain barrier integrity [69] and elevated Phosphodiesterase 4 activity [70], the latter reducing intracellular cAMP levels, thereby removing an endogenous brake on inflammatory signaling cascades.

### 4.3. TMAO: A Molecular Link to Cerebral and Cardiac Damage

Furthermore, dysbiosis leads to an increase in the production of TMA and its derivative, TMAO. TMAO plays a crucial role in amplifying neuroinflammation, which can subsequently cause white matter damage and potentially hemorrhage, a chain of events that may ultimately lead to cardiac dysfunction [26,29]. Specifically, gut dysbiosis, acting through the vagus nerve, enhances the activation of the NLRP3 inflammasome in brain microglia—the main immune cells residing in the brain. This activation results in the release of IL-1β and IL-18, which exacerbate white matter damage and neuroinflammation, possibly leading to intracerebral hemorrhage. The activation of the NLRP3 inflammasome can also worsen white matter injury by disrupting the blood–brain barrier and interfering with nerve regeneration [71].

### 4.4. The Vicious Cycle: Intracerebral Hemorrhage and Cardiac Dysfunction

Intracerebral hemorrhage, in turn, can cause cerebral ischemia (reduced blood flow to the brain) through two mechanisms: increased intracranial pressure, which compresses nearby vessels, and the loss of blood flow from the ruptured vessel itself [72]. Interestingly, in 2022, Hermanns and colleagues demonstrated that cerebral ischemia in mice caused cardiac dysfunction. Their findings suggest that regional neuroinflammation early after cerebral ischemia influences subsequent cardiac dysfunction and described a direct association between neuroinflammation and reduced ejection fraction. Suppression of microglia led to a lower neuroinflammatory signal and was correlated with a preserved cardiac function after cerebral ischemia [73]. Additionally, recent studies reveal a complementary association between heart failure and cerebral collateral flow in large vessel occlusive ischemic stroke. While heart failure is known to cause cerebral hypoperfusion, the relationship between heart failure and the adequacy of collateral flow has not been thoroughly investigated [74,75].

### 4.5. TMAO and Hypertension: A Dual Mechanism of Action

As previously described, the vagus nerve serves as a key intermediary in the complex signaling network that connects intestinal metabolism, including TMAO production [68], with central nervous system responses, thereby contributing to the development of neurological disorders such as stroke, which in turn can cause cardiovascular dysfunction. In addition to the cardiovascular effects outlined previously, Ahmed and colleagues [26] reported that TMAO was strongly associated with cardiovascular disease, even after adjusting for several confounding factors.

The mechanisms linking TMAO to hypertension are multifaceted and operate through parallel pathways. TMAO possesses the ability to cross the blood–brain barrier (BBB), where it induces cognitive dysfunction and neuroinflammation [29].

This neuroinflammatory response is particularly significant because neuroinflammation itself is recognized as a hallmark feature of hypertension [76], creating a pathway through which TMAO can elevate blood pressure via central nervous system activation. Simultaneously, Gao and colleagues demonstrated that this metabolite exerts direct blood pressure-increasing effects by acting on peripheral organs, specifically the heart and kidneys [77]. These organs play crucial roles in blood pressure regulation through their control of cardiac output and fluid balance, respectively. Therefore, TMAO can induce hypertensive effects through two complementary and reinforcing pathways: first, through neuroinflammatory activation in the central nervous system, and second, through direct peripheral action on cardiovascular and renal tissue. This dual mechanism of action makes TMAO a particularly potent contributor to the development and maintenance of elevated blood pressure (Figure 3). The key concepts described in the paragraph are summarized in Figure 4.

### 4.6. The Vagus Nerve and Autonomic Imbalance in Heart Failure

The vagus nerve’s role in TMAO-mediated cardiovascular pathology extends beyond simple signal transmission. Vagal afferent fibers do not cross the epithelial layer [78], and for this reason, they interact with the microbiota by detecting microbial metabolites and conveying this information to the central autonomic network, while efferent fibers facilitate the “cholinergic anti-inflammatory reflex,” which serves to mitigate systemic inflammation and maintain intestinal barrier integrity. In the context of heart failure (HF), reduced vagal tone—a hallmark of autonomic imbalance—may impair this anti-inflammatory feedback mechanism, thereby exacerbating gut permeability and facilitating the translocation of microbial products such as TMA and lipopolysaccharide into the circulation. Consequently, diminished parasympathetic (vagal) activity may potentiate TMAO production and drive inflammatory and fibrotic signaling within the myocardium, thereby establishing a link between dysbiosis and adverse cardiac remodeling, as well as the progression of heart failure [79].

### 4.7. Cellular Mechanisms of TMAO Pathogenicity

At the cellular level, TMAO exerts its pathological effects through multiple molecular mechanisms. Mechanistically, TMAO enters endothelial cells through Endothelial TMAO Transporter-1 and induces the activation of NLRP3, promoting the transition from endothelial to mesenchymal cells [80]. Furthermore, TMAO contributes to atherosclerosis and endothelial dysfunction by promoting macrophage cholesterol accumulation and activating pro-inflammatory pathways such as NF-κB and MAPK, which lead to increased expression of adhesion molecules and cytokines (IL-1β, IL-18, TNF-α) [81,82,83,84,85]. Additionally, TMAO induces oxidative stress and mitochondrial dysfunction through the inhibition of SIRT3 and activation of the NLRP3 inflammasome [86].

## 5. Advanced In Vitro Platforms for Modeling the Gut–Brain–Heart Axis

Emerging clinical questions related to the TMAO-mediated gut–brain–heart axis reveal limitations in classical models: static in vitro systems are too simple, and in vivo models are often too complex to provide the necessary cellular and molecular detail required to isolate specific mechanistic pathways. Advanced in vitro models bridge this methodological gap, moving basic research closer to modeling the complex gut–brain axis and microbiota–gut interactions, often developed within the framework of 3R concepts (Replacement, Reduction, and Refinement of animal models) [87].

Primary cells, immortalized cell lines, and induced pluripotent stem cells (iPSCs) can be cultivated as Next-Generation Testing Tools to better reproduce mammalian physiological conditions, essential for studying inter-organ communication relevant to HF progression [88]. Three main methodological approaches exist based on culture scale, crucial for mimicking the physiological environment of the gut–brain–heart axis:**The Chip Level.** Microfluidic devices containing hair-fine microchannels (tens to hundreds of micrometers) guide and manipulate picoliter-to-milliliter solution volumes [89]. These channels, etched or molded onto substrates like Polydimethylsiloxane or glass, enable high-precision experiments and biological assays at reduced scale.**The Microfluidic Level.** Scaffold-free cellular spheroids (e.g., iPSC-derived) of micrometer dimensions grow three-dimensionally at the base of 1.5 mL tubes with ≤1 mL medium. This system provides physiologically relevant conditions for genetic studies when transgenic models are unavailable, ensuring proper cell–cell contact across all surfaces [90].**The Millifluidic Level.** Coin-sized bioreactors support 2D/3D cell cultures (micro- to millimetric scale) with ~10 mL medium volumes and flow systems mimicking physiological circulation. These platforms investigate laminar flow, shear stress effects, and nonlinear dynamics, proving useful for modeling organ physiopathology, inter-organ communication, and disease conditions [91,92].

One of the first studies exploring gut cell cultures in an advanced bioreactor was published in 2014 [92]. The authors introduced a new dynamic bioreactor, validated using computational fluid dynamic models and pressure analysis, intended to overcome the limitations of traditional static models for studying epithelial permeability. They used fully differentiated Caco-2 cells as a model of the intestinal epithelium, comparing traditional static transwells with the effect of the flow of the medium applied in the bioreactor. They found that flow increased the barrier integrity and tight junction expression of Caco-2 cells compared to static controls. Also, transport across the barrier was increased after flow-induced stimulation, closely replicating the in vivo situation. The successful replication of the intestinal barrier under flow is crucial for accurately modeling the LPS and TMA/TMAO leakage central to the HF pathology.

Recently, Kim and colleagues developed a system to simulate the interaction between gut microbes and human neurons [93]. Employing a gut–brain axis chip containing iPSC-derived human neurons, the authors investigated how metabolites and exosomes produced by gut microbiota affect both neurodevelopmental and neurodegenerative processes. The chip contained channels for intestinal epithelial cells and neural stem cells. Caco-2 cells formed a polarized intestinal layer with in vivo-like morphology, while bacterial co-culture remained stable without overgrowth. The authors were able to demonstrate that microbial metabolites and exosomes promoted neural differentiation and synaptic protein expression in iPSC-derived neurons.

Horvath et al. [94] proposed a protocol including bacterial cultures, organoids, and in vivo samples which would enable studies on the connections between microbial colonization and intestinal and brain neurotransmitters. This strategy would allow for a comprehensive evaluation of metabolites in an in vitro system represented by organoids and in vivo mouse model systems.

In 2019, the group of Giordano et al. [95] introduced MINERVA (MIcrobiota-Gut-BraiN EngineeRed platform to eVAluate intestinal microflora impact on brain functionality). This is a multi-organ-on-a-chip platform specifically designed to investigate the influence of the microbiota on neurodegeneration. The device facilitates the co-cultivation and communication between cells representing the microbiota, the gut epithelium, the immune system, and the blood–brain barrier all integrated onto a single chip. The future application would be to assess drug therapeutic strategies in a context of personalized medicine approach.

Hall and colleagues [96] discussed the use of iPSCs in gut–brain axis chips and the challenges to be faced, and Raimondi et al. summarized several organ-on-a-chip models to study the microbiota-gut–brain axis and its involvement in neurological diseases [97].

The primary goal, therefore, is to develop a comprehensive gut–brain–heart multi-organ system. This involves placing cardiac cells, which have already been successfully cultivated and studied in flow-based bioreactors [91,98], in communication with the described gut–brain axis chip devices. This integration would allow the reproduction, with unprecedented detail, of the complex cross-talk that exists between these systems. This approach represents the application possible in the immediate future: assessing drug therapeutic strategies and personalized medicine approaches specifically targeting the TMAO pathway and its downstream effects on the heart.

## 6. Therapeutic Directions

Diet influences TMAO through both food type and total energy intake [99,100,101]. Beyond lifestyle changes, various pharmacological and bioactive strategies are emerging to mitigate the deleterious effects of the microbiota-derived metabolite TMAO. Current evidence suggests that conventional cardiovascular drugs, such as statins and loop diuretics, significantly interact with TMAO pathways, either by reducing its production or by influencing its renal excretion. Furthermore, the use of targeted nutraceuticals offers a promising non-pharmacological approach to reshape the gut microbiota and attenuate TMAO-induced vascular and metabolic damage. The following sections detail how these diverse interventions—ranging from lipid-lowering therapies to polyphenol-rich extracts—can be integrated into heart failure management to improve patient outcomes and risk stratification.

### Pharmacological Interventions

Statins:

Statin therapy reduces TMAO. Daniel Y. Li et al., in a major study, showed that therapy with atorvastatin (80 mg) or atorvastatin 10 mg + ezetimibe 10 mg significantly reduces TMAO [102].

Rosuvastatin also reduces the ability of the intestinal flora to produce TMAO [103].

In patients treated with statins, therapeutic efficacy, expressed as the HDL/LDL ratio, correlates with TMAO levels. TMAO analysis allows therapy to be modulated by improving risk stratification and reducing the incidence of major cardiovascular events.

Diuretics of the loop of Henle:

Loop of Henle diuretics, such as furosemide, have been and remain fundamental in the treatment of heart failure. However, prolonged use of these diuretics can cause worsening of kidney function and increased mortality. Li D.Y. et al. demonstrated a direct relationship between loop of Henle diuretics, TMAO, and major cardiovascular events [104]. This is because loop of Henle diuretics compete with the mechanism of TMAO excretion.

Nutraceuticals:

Nutraceuticals such as resveratrol, catechins, quercetin, and Taurisolo^®^ show promising activity in counteracting TMAO-related metabolic and vascular damage. Resveratrol, a polyphenol abundant in grapes and berries, reduces circulating TMAO levels and promotes hepatic bile acid synthesis by reshaping the gut microbiota while also activating sirtuins and mitigating mitochondrial inhibition and inflammasome activation [105].

Catechins and quercetin—widely present in fruits and vegetables—exert potent antioxidant, anti-inflammatory, and vasoprotective effects; notably, quercetin also lowers TMAO and attenuates its hepatotoxicity [106].

Taurisolo^®^, a standardized Aglianico grape pomace extract rich in these polyphenols, has demonstrated neuroprotective and endothelial-protective effects in preclinical models and significantly reduces TMAO through multiple cellular and metabolic pathways (Table 2) [107,108].

## 7. Conclusions and Future Therapeutic Directions

In conclusion, this complex pathophysiological cascade demonstrates how gut dysbiosis can initiate a devastating chain reaction through vagal signaling, leading to neuroinflammation, cerebral damage, and subsequent cardiac dysfunction. The bidirectional nature of this gut–brain–heart axis creates a vicious cycle where cardiac dysfunction can further compromise cerebral perfusion, potentially exacerbating neuroinflammation and perpetuating the pathological process. Metabolites like TMAO serve as molecular messengers in this intricate network, simultaneously affecting both neural and cardiovascular systems through multiple pathways. Understanding the impact of TMAO and associated conditions like stroke and hypertension is crucial, as they exacerbate chronic heart conditions. Atrial fibrillation and ventricular arrhythmias are well-known causes and complications of heart failure, further underscoring the severity of these interconnected cardiovascular events. A deeper comprehension of these networked mechanisms opens new therapeutic avenues targeting the vagus nerve, gut microbiota modulation, or anti-inflammatory strategies to break this pathological cycle.

Future translational research must intensify its focus on dissecting the complex, multidirectional gut–brain–heart axis dysfunction. This crucial endeavor necessitates the robust application of cutting-edge research methodologies, particularly advanced in vitro tools such as organ-on-a-chip models and complex co-culture systems that faithfully replicate in vivo physiological conditions. The primary scientific objective must be the precise identification of early, predictive biomarkers. These markers, which could include specific microbial metabolites, inflammatory mediators, or neurohormonal signals, are essential for recognizing the incipient stages of dysfunction long before overt symptoms manifest.

Simultaneously, this research effort must transition into the development of highly targeted interventions. These strategies should move beyond broad-spectrum approaches to focus specifically on restoring eubiosis and correcting the pathological signaling pathways initiated by gut dysbiosis. The ultimate and critical goal is to create effective prophylactic and therapeutic measures capable of preventing the progression from a disturbed gut microbiome and associated systemic inflammation with severe, life-threatening cardiovascular complications, thereby significantly improving patient prognosis and public health outcomes.

## Figures and Tables

**Figure 1 ijms-27-00203-f001:**
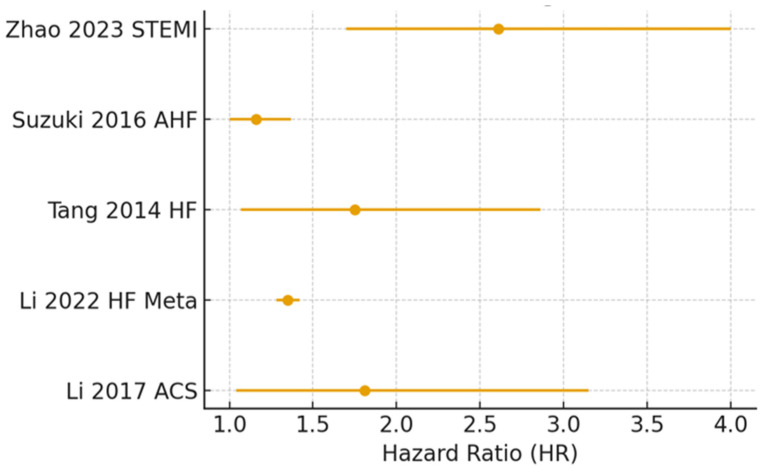
Forest plot of prognostic studies on TMAO visually representing effect estimates from the individual studies presented in the table regarding the association between elevated TMAO levels and major clinical outcomes, such as mortality or adverse events, in heart failure patients [39,43,44,55,56].

**Figure 2 ijms-27-00203-f002:**
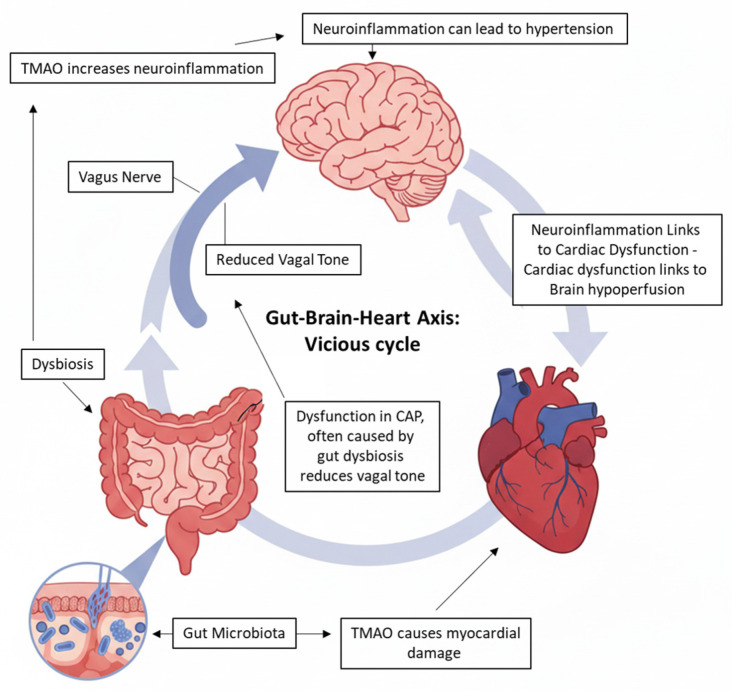
Gut dysbiosis initiates a devastating vagal-mediated chain: neuroinflammation leads to cerebral damage and subsequent cardiac dysfunction. This bidirectional gut–brain–heart axis forms a vicious cycle, exacerbated by metabolites like TMAO, perpetuating pathology.

**Figure 3 ijms-27-00203-f003:**
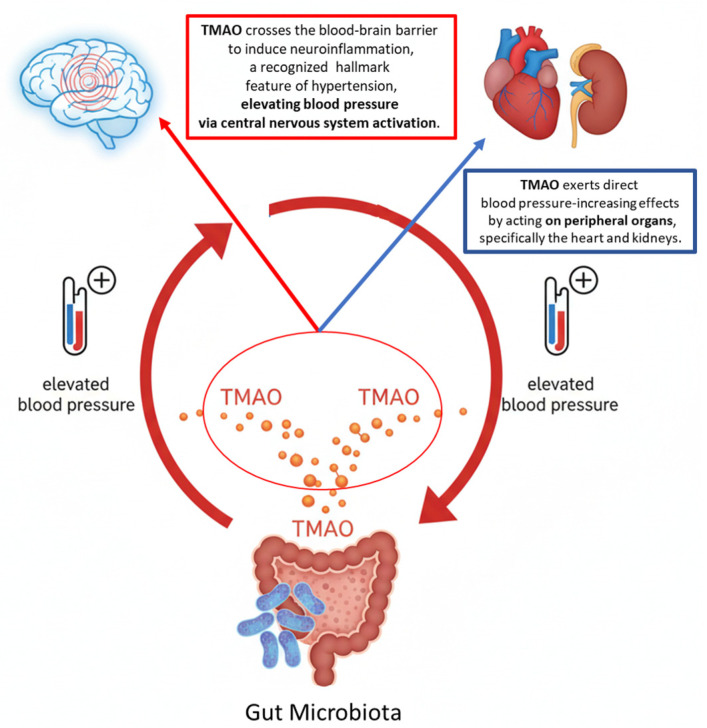
The vagus nerve links intestinal TMAO production to the central nervous system. TMAO drives hypertension through a dual mechanism: inducing neuroinflammation by crossing the blood–brain barrier and exerting direct blood pressure-increasing effects on peripheral organs like the heart and kidneys.

**Figure 4 ijms-27-00203-f004:**
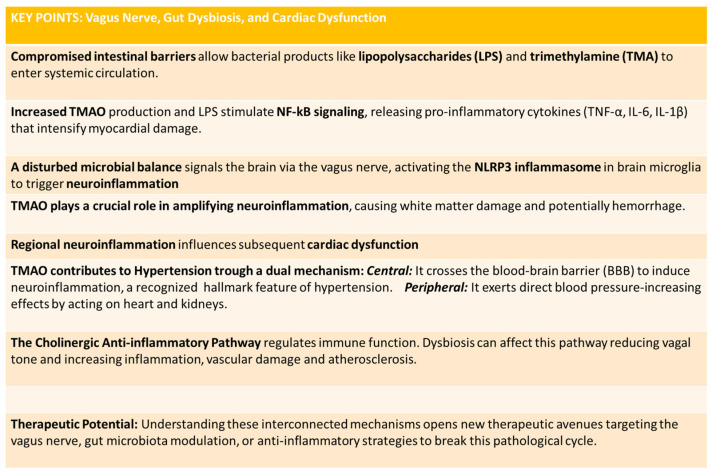
The key concepts described in Section 4.2.

**Table 1 ijms-27-00203-t001:** Comparative table of key clinical studies on TMAO’s prognostic value.

Study/Year	Population	Setting	Follow-Up	Main Findings
Li et al., 2017 [39]	N. pts. = 530	Acute coronary syndrome	30 day–7 year	TMAO independently predicts MACEs and mortality
Li et al., 2022 [55]	Meta-analysis 13,425 pts.	HF	Varied	Higher TMAO predicts MACEs and all-cause mortality
Tang et al., 2014 [44]	N. pts. = 720	Chronic HF	5 year	High TMAO predicts long-term mortality
Suzuki et al., 2016 [43]	N. pts. = 972	Acute HF	1 year	TMAO predicts in-hospital and 1 year mortality/HF readmission
Zhao et al., 2023 [56]	N. pts. = 1004	ST-elevation myocardial infarction–percutaneous coronary intervention	1 year	TMAO predicts MACEs independent of risk factors

**Table 2 ijms-27-00203-t002:** Therapeutic directions—a summary.

Therapeutic Area	Intervention/Key Concept	Mechanism or Clinical Relevance	Key References
Dietary Interventions	Diet composition (fiber, animal proteins, dietary patterns)	Diet modulates TMAO production; fiber reduces postprandial TMAO peaks.	Haas et al. 2025 [99] Thomas et al. 2021 [100] Krishnan et al. 2022 [101]
Statins	Atorvastatin, Atorvastatin + Ezetimibe, Rosuvastatin	Statins reduce TMAO levels; improved CV risk stratification.	Li DY et al. 2018 [102] Kummen et al. 2020 [103]
Loop Diuretics	Furosemide and other loop diuretics	Compete with renal TMAO excretion → increased circulating TMAO → higher CV risk.	Li D.Y. et al. 2021 [104]
Nutraceuticals—Resveratrol	Resveratrol	Reduces TMAO; enhances bile acid synthesis; modulates microbiota; activates SIRT pathways.	Chen M et al. 2016 [105]
Nutraceuticals—Catechins	Catechin polyphenols	Exert antioxidant and vasoprotective effects.	Chen M et al. 2016 [105] Annunziata et al. [107] Amato et al. 2024 [108]
Nutraceuticals—Quercetin	Quercetin	Lowers TMAO; reduces hepatotoxicity; exerts anti-inflammatory and antioxidant effects.	Zhang et al. 2023 [106]
Nutraceuticals—Taurisolo^®^	Aglianico grape extract	Exert neuroprotective and endothelial-protective effects; reduces TMAO in preclinical studies.	Annunziata et al. [107] Amato et al. 2024 [108]

## Data Availability

No new data were created or analyzed in this study. Data sharing is not applicable to this article.

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
