# Peer review of "Interplay Among Gut Microbiota-Derived TMAO, Autonomic Nervous System Dysfunction, and Heart Failure Progression"

_ijms, 2025, doi:10.3390/ijms27010203_

Round 1

Reviewer 1 Report

Comments and Suggestions for Authors

This review systematically summarizes the relationship between Trimethylamine N-oxide (TMAO), autonomic nervous system dysfunction, and heart failure. There are several issues that need to be addressed to improve the manuscript.

  1. The language should be improved in the manuscript.
  2. The manuscript does not adhere to standard conventions for abbreviations. Many abbreviations are used without being defined upon their first occurrence. Furthermore, abbreviations that appear only once should be spelled out in full. Please note that the abstract and the main text are considered separate sections for the purpose of defining abbreviations.
  3. The keyword list requires revision. Firstly, some terms are not central to the manuscript's theme. Secondly, the capitalization should be made consistent. The established abbreviation "TMAO" is appropriately used and does not need to be spelled out.
  4. The Introduction lacks a clear rationale for the study. It should be revised to explicitly articulate the specific objective of investigating the interrelationship among TMAO, autonomic nervous system dysfunction, and heart failure.
  5. The purpose of the table on page 2 is unclear, as it does not appear to provide the literature.
  6. The manuscript should be carefully reviewed, as several statements lack supporting citations. For instance, the claims made in lines 117-124 require appropriate references.
  7. The logical flow of the manuscript needs to be improved to enhance the overall clarity and narrative coherence.
  8. The section describing "Advances in vitro models to study gut-brain axis" appears tangential to the central theme of the manuscript. Please clarify its specific relevance to the main topic or consider revising this part.

Author Response

This review systematically summarizes the relationship between Trimethylamine N-oxide (TMAO), autonomic nervous system dysfunction, and heart failure. There are several issues that need to be addressed to improve the manuscript.

  1. The language should be improved in the manuscript.

We thank the Reviewer for this suggestion. Language has been improved.

  1. The manuscript does not adhere to standard conventions for abbreviations. Many abbreviations are used without being defined upon their first occurrence. Furthermore, abbreviations that appear only once should be spelled out in full. Please note that the abstract and the main text are considered separate sections for the purpose of defining abbreviations.

The comment is appreciated since highlighted inaccuracies that escaped an initial review. Abbreviations have been corrected in both the table and the text, in accordance with the suggestion.

  1. The keyword list requires revision. Firstly, some terms are not central to the manuscript's theme. Secondly, the capitalization should be made consistent. The established abbreviation "TMAO" is appropriately used and does not need to be spelled out.

Thank you for the suggestion. Keywords have been revised as required.

  1. The Introduction lacks a clear rationale for the study. It should be revised to explicitly articulate the specific objective of investigating the interrelationship among TMAO, autonomic nervous system dysfunction, and heart failure.

We thanks the reviewer for highlighting this weakness and we hope that the new paragraph and references will comply with her/his expectations.

  1. The purpose of the table on page 2 is unclear, as it does not appear to provide the literature.

Thank you for the comment. The Key Point figure is designed to highlight the main concept of the review, without reference citation (an example is the same object in the published study: Calvillo, L., Gironacci, M.M., Crotti, L. et al. Neuroimmune crosstalk in the pathophysiology of hypertension. Nature Rev Cardiol 16, 476–490 (2019). https://doi.org/10.1038/s41569-019-0178-1). Nevertheless, we agree that the figure is not clear enough and is too extensive. A more concise Key Point section is now provided.

  1. The manuscript should be carefully reviewed, as several statements lack supporting citations. For instance, the claims made in lines 117-124 require appropriate references.

We agree with the comment, thank you. New references were added to make the sentences more appreciable.

  1. The logical flow of the manuscript needs to be improved to enhance the overall clarity and narrative coherence.

We thank the Reviewer for this suggestion. By adding the new paragraphs we also implemented the overflow flow of information and we hope that the reading of this paper is now more fluent.

  1. The section describing "Advances in vitro models to study gut-brain axis" appears tangential to the central theme of the manuscript. Please clarify its specific relevance to the main topic or consider revising this part.

Thank you for the insightful comment. We agree that the relevance of the Advanced in vitro models section requires clearer justification. This section is central to the manuscript's thesis because the pathology we describe—the complex, bidirectional interplay among Gut Dysbiosis, Vagal Signaling, Neuroinflammation, and Cardiac Dysfunction—cannot be fully recapitulated or investigated using traditional static in vitro or overly complex in vivo models.

Advanced platforms (Chip/Fluidic Levels) are necessary methodological tools to:

  1. Isolate and study the mechanism of TMAO production and its transport across the gut and blood-brain barriers.
  2. Model the specific cross-talk between the gut epithelium, the autonomic nervous system components (neurons), and cardiac cells (as highlighted in the proposed future goal).

We have revised the section title and text to explicitly link these advanced models to the necessity of reproducing the multi-organ pathogenesis of heart failure (HF) discussed in the Abstract, thereby reinforcing its specific relevance to our manuscript's core theme.

Reviewer 2 Report

Comments and Suggestions for Authors

This comprehensive review article presents a timely and integrative overview of the complex role of TMAO within the gut-brain-heart axis, with a specific focus on autonomic nervous system dysfunction and heart failure progression. I have some comments:

1) The manuscript would benefit from significant structural streamlining. The "KEY POINTS" section in the introduction is extensive and repetitive, pre-empting much of the detail that is more effectively explained in later dedicated sections. It is recommended to condense the "KEY POINTS" into a concise, bulleted list of the core hypotheses or to integrate its concepts more seamlessly into the narrative flow of the introduction. Either move this block to the end of the article, before the conclusion.

2) The review reads largely as a summation of supportive evidence. To enhance its scholarly impact, it should more explicitly acknowledge controversies, knowledge gaps, and contradictory findings in the field.

3) Section "ANS in HF" provides excellent historical context on autonomic nervous system modulation in heart failure. However, the direct link to TMAO in this specific section is tenuous. A concluding paragraph explicitly connecting the established role of sympathetic overdrive/vagal withdrawal to the potential mediating role of TMAO would better integrate this section with the paper's central theme.

4) While the association between TMAO and cardiovascular diseases is strong, is it truly causative, or merely a marker of renal impairment or overall health status?

5) The discussion on therapeutic interventions is mentioned but not critically evaluated. What is the current clinical evidence for their efficacy in reducing TMAO and improving hard cardiovascular outcomes?

6) A comparative table summarizing the key clinical studies and their findings regarding TMAO's prognostic value would greatly strengthen the clinical sections.

7) A thorough proofreading by a native English speaker is recommended to improve readability and professionalism.

Author Response

This comprehensive review article presents a timely and integrative overview of the complex role of TMAO within the gut-brain-heart axis, with a specific focus on autonomic nervous system dysfunction and heart failure progression. I have some comments:

1) The manuscript would benefit from significant structural streamlining. The "KEY POINTS" section in the introduction is extensive and repetitive, pre-empting much of the detail that is more effectively explained in later dedicated sections. It is recommended to condense the "KEY POINTS" into a concise, bulleted list of the core hypotheses or to integrate its concepts more seamlessly into the narrative flow of the introduction. Either move this block to the end of the article, before the conclusion.

Thank you for this important comment. The "KEY POINTS" are now more concise as suggested.

2) The review reads largely as a summation of supportive evidence. To enhance its scholarly impact, it should more explicitly acknowledge controversies, knowledge gaps, and contradictory findings in the field.

We are grateful for this suggestion which allowed us to improve the manuscript. We now have addressed the actual controversies and gap in the Conclusion section.

3) Section "ANS in HF" provides excellent historical context on autonomic nervous system modulation in heart failure. However, the direct link to TMAO in this specific section is tenuous. A concluding paragraph explicitly connecting the established role of sympathetic overdrive/vagal withdrawal to the potential mediating role of TMAO would better integrate this section with the paper's central theme.

We thank the reviewer for this important criticism. Making clear the unfavourable process determined by the interaction between microbiota, ANS, inflammation and negative post MI remodeling is indeed a goal of this review. We hope that the additions do indeed address the reviewer’s expectation

4) While the association between TMAO and cardiovascular diseases is strong, is it truly causative, or merely a marker of renal impairment or overall health status?

Thank you for highlighting this important point. A clarification on the relation between TMAO and Kidney has now present in the text (from line 198). 

5) The discussion on therapeutic interventions is mentioned but not critically evaluated. What is the current clinical evidence for their efficacy in reducing TMAO and improving hard cardiovascular outcomes?

We appreciate the Reviewer’s suggestion, we are agree that the therapeutic intervention had to be further discussed. A new section entitled Therapeutic Directions is now present in the manuscript (line 435)

6) A comparative table summarizing the key clinical studies and their findings regarding TMAO's prognostic value would greatly strengthen the clinical sections.

We thank the Reviewer for this important suggestion which contributed to improve the clinical section. A comparative Table is now present, as well as a new figure describing a Forest Plot of TMAO prognostic studies (Table 1 and new Figure 1).  

7) A thorough proofreading by a native English speaker is recommended to improve readability and professionalism.

We thank the Reviewer for this suggestion. Language has now been improved.

Round 2

Reviewer 1 Report

Comments and Suggestions for Authors

The manuscript has been revised in response to the reviewers' comments. Nonetheless, the following issues have been identified.

  1. The abstract seems to be introducing the background of the paper, but lacks an explanation of why this review is conducted and what this paper covers.
  2. Issues with abbreviation formatting persist. Abbreviations are defined at their first mention in the Abstract or the main text. Any term that is abbreviated but used only once in the entire text should be written in full. For example, lines 28, 48, 58, 113, 146, etc. Also, please verify whether the full names of the abbreviations are correct. For example, lines 78, 79, etc.
  3. The key words were not modified as suggested.
  4. The figure on line 67 serves a limited purpose within the context of this manuscript and is recommended for removal. Please note that this journal's format does not include a graphical abstract. If a visual summary is desired, it is recommended to add a standardized summary figure (e.g., a schematic diagram) at the end of the article. Note that a table should not be formatted or submitted as a figure.
  5. It is admitted that the section "ANS in HF" has a certain effect on readers' understanding, but the logic of this section is not very smooth and the key points are not very prominent. The discussion of the historical development of ANS in HF deviates from the paper's core theme. It is recommended to reconstruct this section to concisely explain why ANS dysfunction is critical in HF pathophysiology and how it exerts its effects. Subsequently, given that the following content centers on vagus nerve, the narrative should pivot to specifically elaborate on the role of the vagus nerve in HF.
  6. Lines 117-119, this sentence does not match the content of this part.
  7. Lines 136-143, the two paragraphs seem to have little significance for this part.
  8. Lines 211-212, figures and tables need certain descriptions in the text, not just a simple sentence.
  9. The "Therapeutic Directions" section lacks depth and should be expanded. A summary table is also needed.

Author Response

The manuscript has been revised in response to the reviewers' comments. Nonetheless, the following issues have been identified.

  1. The abstract seems to be introducing the background of the paper, but lacks an explanation of why this review is conducted and what this paper covers.

We understand the issue and added a new paragraph at the end of the abstract

  1. Issues with abbreviation formatting persist. Abbreviations are defined at their first mention in the Abstract or the main text. Any term that is abbreviated but used only once in the entire text should be written in full. For example, lines 28, 48, 58, 113, 146, etc. Also, please verify whether the full names of the abbreviations are correct. For example, lines 78, 79, etc.

Abbreviations have been corrected as required.

  1. The key words were not modified as suggested.

Keywords have been modified accordingly

  1. The figure on line 67 serves a limited purpose within the context of this manuscript and is recommended for removal. Please note that this journal's format does not include a graphical abstract. If a visual summary is desired, it is recommended to add a standardized summary figure (e.g., a schematic diagram) at the end of the article. Note that a table should not be formatted or submitted as a figure.

 We believe the content of this figure is very helpful for the clear understanding of the paper and we would like to keep it in a bullet point table to match the journal policy.

  1. It is admitted that the section "ANS in HF" has a certain effect on readers' understanding, but the logic of this section is not very smooth and the key points are not very prominent. The discussion of the historical development of ANS in HF deviates from the paper's core theme. It is recommended to reconstruct this section to concisely explain why ANS dysfunction is critical in HF pathophysiology and how it exerts its effects. Subsequently, given that the following content centers on vagus nerve, the narrative should pivot to specifically elaborate on the role of the vagus nerve in HF.

We appreciated the reviewer’s comment and changed the paragraph accordingly

  1. Lines 117-119, this sentence does not match the content of this part.

This has been changed while revising the entire paragraph

  1. Lines 136-143, the two paragraphs seem to have little significance for this part.

This has been changed while addressing issue 5

  1. Lines 211-212, figures and tables need certain descriptions in the text, not just a simple sentence.

We understand the issue and added legends

The "Therapeutic Directions" section lacks depth and should be expanded. A summary table is also needed.

We added a summary table as required

Reviewer 2 Report

Comments and Suggestions for Authors

I approve revised paper.

Author Response

(The authors gave the same response as above.)

Round 3

Reviewer 1 Report

Comments and Suggestions for Authors

While the authors have addressed some of the reviewers' points, there are a few issues with the manuscript.

1. The abstract is rather long and needs some deletions. The abstract only needs one paragraph. The logic of the abstract still needs to be strengthened.
2. Line 74, the standardized figure needs titles in the text and should also be described in the main body.
3. Lines 119-120, “Overall, there is a documented loop linking microbiota, TMAO inflammation sympathetic activation to cardiovascular disease risk and post MI progression into HF” This part does not cover any related content and should be deleted.
4. Lines 148-149, “please see the corrected article at https://dmd.aspetjournals.org/ar-148 ticle/S0090-9556(24)09966-5/fulltext” Some similar content exists after the citation numbers in the text. By checking the content, it is found that it is consistent with the corresponding paper of the citation numbers. If there is a replacement, this method is not standardized.
5. Lines 225-227, 230-233, The description of Figure 1 and Table 1 should be placed after the sentence in line 21.
6. In Table 1, “y” should use "year".
7. Line 291, “Fig.1” needs to be modified.
8. Line 348, “Fig.2” needs to be modified.
9. Line 445-457, the first paragraph of this section should be a summary of current clinical or experimental treatments and their mechanisms, followed by a separate description of treatments.
10. Line 492, The title of the table needs to be numbered.

Author Response

While the authors have addressed some of the reviewers' points, there are a few issues with the manuscript.

  1. The abstract is rather long and needs some deletions. The abstract only needs one paragraph. The logic of the abstract still needs to be strengthened.

Done

  1. Line 74, the standardized figure needs titles in the text and should also be described in the main body.

The picture at the beginning of the article represents keypoints, as published, for example, on Nature Reviews in Cardiology  [1]. Nevertheless, in order to meet Reviewer’s requirement, the picture has been transformed in Figure and moved to line 340. A link in the text has been provided.

  1. Lines 119-120, “Overall, there is a documented loop linking microbiota, TMAO inflammation sympathetic activation to cardiovascular disease risk and post MI progression into HF” This part does not cover any related content and should be deleted.

Done

  1. Lines 148-149, “please see the corrected article at https://dmd.aspetjournals.org/ar-148 ticle/S0090-9556(24)09966-5/fulltext” Some similar content exists after the citation numbers in the text. By checking the content, it is found that it is consistent with the corresponding paper of the citation numbers. If there is a replacement, this method is not standardized.

Done

  1. 5. Lines 225-227, 230-233, The description of Figure 1 and Table 1 should be placed after the sentence in line 21.

Line 21 is in the abstract, we assume that the Reviewer meant the lines before 225. Done as required.

  1. In Table 1, “y” should use "year".

Done

  1. Line 291, “Fig.1” needs to be modified.

Done

  1. Line 348, “Fig.2” needs to be modified.

Done

  1. 9. Line 445-457, the first paragraph of this section should be a summary of current clinical or experimental treatments and their mechanisms, followed by a separate description of treatments.

Done

  1. Line 492, The title of the table needs to be numbered.

Done

[1]          Calvillo L, Gironacci MM, Crotti L, Meroni PL, Parati G. Neuroimmune crosstalk in the pathophysiology of hypertension. Nat Rev Cardiol 2019;16:476–90. https://doi.org/10.1038/s41569-019-0178-1.
